# Improving desirable agronomic traits of M2 lines on fourteen Ethiopian Sesame (*Sesamum indicum* L.) genotypes using Ethyl Methane Sulphonate (EMS)

**Micheale Yifter Weldemichael**[1]*, **Tesfaye Dissasa Bitima**[2], **Getachew Tafere Abrha**[1], **Kalkidan Tesfu**[2], **Hailay Mehari Gebremedhn**[1], **Abraha Birhan Kassa**[1], **Yirga Belay Kindeya**[3], **Mohammed Mebrahtu Mossa**[1]

**1** Mekelle University, Mekelle, Tigrai, Ethiopia, **2** National Agricultural Biotechnology Research Center, Holleta, Ethiopia, **3** Humera Agricultural Research Center, Humera, Tigrai, Ethiopia

* y.mickye@gmail.com, micheale.yifter1@mu.edu.et

## Abstract

Sesame is an important oilseed crop cultivated in Ethiopia as a cash crop for small holder farmers. However, low yield is one of the main constraints of its cultivation. Boosting and sustaining production of sesame is thus timely to achieve the global oil demand. This study was, therefore, aimed at identifying mutant genotypes targeted to produce better agronomic traits of M2 lines on fourteen Ethiopian sesame genotypes through seed treatment with chemical mutagens. EMS was used as a chemical mutagen to treat the fourteen sesame genotypes. Quantitative and qualitative data were recorded and analyzed using analysis of variance with GenStat 16 software. Post-ANOVA mean comparisons were made using Duncan's Multiple Range Test (p≤ 0.01). Statistically significant phenotypic changes were observed in both quantitative and qualitative agronomic traits of the M2 lines. All mutant genotypes generated by EMS treatment showed a highly significant variation for the measured quantitative traits, except for the traits LBL and LTL. On the other hand, EMS-treated genotypes showed a significant change for the qualitative traits, except for PGT, BP, SSCS, LC, LH and LA traits. Mutated Baha Necho, Setit 3, and Zeri Tesfay showed the most promising changes in desirable agronomic traits. To the best of our knowledge, this study represents the first report on the treatment of sesame seeds with EMS to generate desirable agronomic traits in Ethiopian sesame genotypes. These findings would deliver an insight into the genetic characteristics and variability of important sesame agronomic traits. Besides, the findings set up a foundation for future genomic studies in sesame agronomic traits, which would serve as genetic resources for sesame improvement.

## Introduction

Sesame (*Sesamum indicum* L., 2n = 26) is one of the ancient oilseed crops widely cultivated, mainly in the arid and semi-arid regions of Africa, Asia and South America, as a source of

**Funding:** This research was supported by the Ethiopian Institute of Agricultural Research (grant number EIAR/025/2019). The funders had no role in study design, data collection and analysis, decision to publish, or preparation of the manuscript.

**Competing interests:** The authors have declared that no competing interests exist.

high-quality oil and edible seeds [1, 2]. Sesame seed has a high oil quality and quantity, nutrition, cosmetic and pharmaceutical uses [3]. Sesame seeds are known to be rich in carbohydrate, protein, oil, and dietary fiber [4]. Besides, sesame seed has abundant applications for nutritional, industrial and pharmaceutical uses due to its high oil quality, high oil content, and strong resistance to oxidation, and hence is referred as the 'queen of oilseeds' [1, 5]. Compared to the seeds of other main oil crops, such as peanut (*Arachis hypogea*), rapeseed (*Brassica napus*), soybean (*Glycine max*), and olive (*Olea europaea*), sesame seeds not only have the highest oil content, but also are rich in vitamins and minerals [6]. Moreover, sesame seed is an excellent source of vegetable oil which is rich in polyunsaturated fatty acids, phytosterols, tocopherols, and unique classes of lignans such as sesamolin and sesamin, all of which are known as important compounds for human health [1, 7]. Sesamin is endowed with anti-cancer property [8, 9], whereas sesamol has a powerful functional ingredient for treating cardiovascular diseases [10]. In addition, sesame seed serves as a major component in the production of cosmetics, soaps, perfumes, pharmaceutical products, and insecticides [11]. As a result of such wider economical, industrial, nutritional and medicinal significances, the demand for sesame oil consumption is forecasted to reach about 220.46 million tons by 2030 [12]. Improving the production and productivity of this oilseed crop, *Sesamum indicum* L., is hence prominent to alleviate the alarmingly growing demands of oil. However, the production and productivity of sesame are challenged by numerous factors including early flowering, shattering, indeterminate growth habit, and branching as well as diseases and insect-pests such as bacterial blight, Cercospora leaf spot, Fusarium wilt, phyllody, webworm and aphids as major ones, which are causing significant yield losses every year.

Sesame yield is highly affected by numerous factors including plant height, growth habit, branching habit, internode length, and flowers per leaf axial [13–17]. The low yield potential of sesame, as compared to other oilseed crops, is mainly due to its indeterminate growth habit, which is characterized by continuous flowering and non-uniform ripening of capsules [14]. In addition, plant height and internode length are among the most important agronomic traits for sesame production [18], which lead to low yielding capacity, low harvest index, lodging, and susceptibility to various biotic and abiotic stresses [19]. Hence, enhancing sesame production and productivity through the improvement of these traits is among the significant aims of sesame breeding programs [20, 21]. As a result, various chemical mutagens such as EMS, diethyl sulfate, colchicine, sodium azide, and hydrazine hydrate have been used in different crops [22, 23].

Chemical mutagens, chemical agents that permanently change genetic material, are vital to significantly reduce plant height and enhance lodging resistance so as to increase sesame production [22]. Induced mutation using EMS is an easy and inexpensive alternative for breeding of crops to improve yield, early maturity, quality traits, as well as to confer resistance against biotic and abiotic stresses [23]. Seeds treated with EMS enhanced sesame yield through the growth of cultivars with determinate growth habit, synchronous maturity and earliness, improved pod shattering resistance, resistance to diseases, larger seed size, male sterility, higher oil content and modified fatty acid composition [24]. Furthermore, a gene *SiDt* was detected as a target gene for conferring the determinate trait in sesame cultivar using genetic mapping and genomic association studies [13]. Similarly, *Sidwf1* was reported as a gene controlling short internode length and dwarfing trait in *dw607* mutant sesame [16]. Another candidate gene (*SiBH*) for branching habit, an important trait in sesame affecting cultivation practice and mechanical harvesting, was reported to facilitate mechanical harvesting without compromising seed yield [17]. More recently, the work of Weldemichael et al. [25] revealed the effects of sodium azide on various quantitative and qualitative stem traits of M2 lines on fourteen Ethiopian sesame genotypes. However, an intensive mutation breeding using EMS

targeting key functional agronomic traits for the Ethiopian sesame genotypes is still demanding. Therefore, it is vital to develop new mutants using EMS, which can accelerate the breeding program and improve sesame genotypes. The main objective of this study was to use EMS to induce desirable agronomic traits in 14 Ethiopian sesame genotypes, with the aim of accelerating breeding programs and improving the crop's productivity.

## Materials and methods

### Experimental site

In this study, both laboratory and field experiments have been conducted. The laboratory experiment was carried out at Tigrai Biotechnology Center, Pvt. Ltd. Co. in Mekelle city, Tigrai region, northern Ethiopia (Lat.: 13˚ 30′ 0″ N; Long.: 39˚ 28′ 11″ E; Alt. 2,080 masl). In addition, the field evaluation of the fourteen sesame genotypes was undertaken in three environments, namely Humera (14˚15′ N, 36˚37′ E), Kebabo (13˚36′ N, 36˚41′ E), and Sheraro (14˚24' N, 37˚ 45' E) for two rainy seasons, 2019 and 2020 (Fig 1). The three sites have similar climatic conditions and represent the major sesame production areas in Ethiopia [26].

### Collection of seeds and sterilization

Sesame seeds of fourteen genotypes comprising four landraces, an introduced variety from Israel and nine varieties released from different research centers in Ethiopia, which were acquired from Humera Agricultural research center (HuARC), Tigrai, Ethiopia, were used in this study (Table 1). Disease-free, dry and normal shaped seeds were used for the experiment. The seeds were soaked in 3% teepol detergent solution for 5 min after being washed with running tap water for 20 min and rinsed with distilled water. 70% ethanol was used to disinfect the seeds for 45 sec at room temperature and rinsed with sterile distilled water three times. Sulfuric acid and glacial acetic acid were used to treat the seeds and soaked in sterile distilled water for 16 hrs [25, 27].

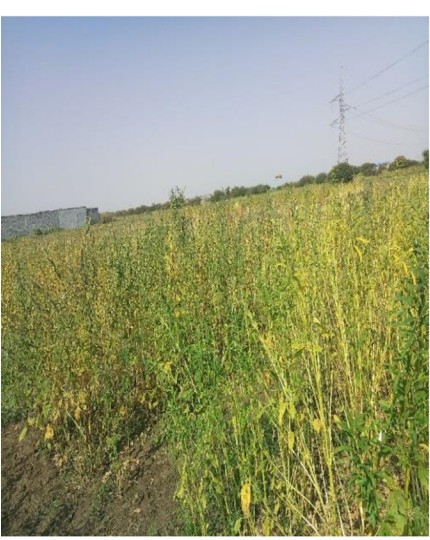 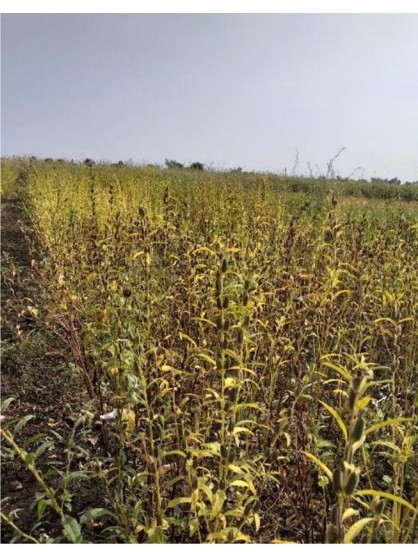 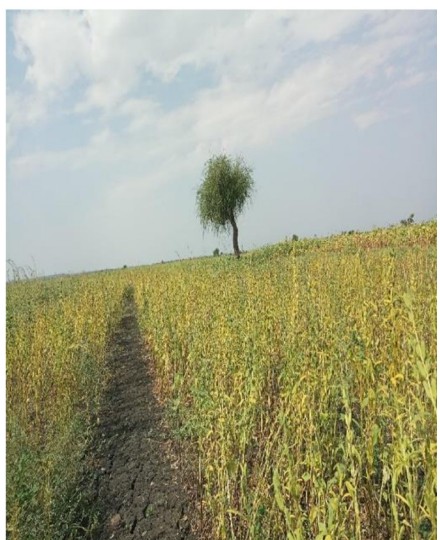

| A. Humera | B. Kebabo | C. Sheraro |

**Fig 1. Sesame genotypes with different agronomic traits.** The image was taken from fourteen sesame genotypes planted in three locations. A. Humera, B. Kebabo, C. Sheraro.

**Table 1. List of 14 sesame genotypes with their pedigree, altitude, average gain yield, oil content and days to maturity.**

| Genotype name | Year of release | Local/ improved | Pedigree name | Average grain yield (Kgha$^{-1}$) | Days to maturity | Altitude (m.a.s.l) * | Oil content (%) | Released by |
|---|---|---|---|---|---|---|---|---|
| Baha Necho | 2016 | Improved | Acc-EW-012 (5) | 1200 | 114–129 | 560–1650 | 52 | HU |
| Baha Zeyit | 2016 | Improved | Acc-EW-012 (3) | 1300 | 113–134 | 560–1650 | 56 | HU |
| Humera 1 | 2011 | Improved | Acc038 sel 1 | 590–900 | 90–110 | 600–1100 | 54.56 | Humera ARC |
| ACC44 | 2013 | Improved | Acc0047 | 700–800 | 105–120 | - | 50.4 | Sirinka ARC |
| Setit 1 | 2011 | Improved | Col sel p#1 | 620–1000 | 80–90 | 600–800 | 52.54 | Humera ARC |
| Borkena | 2007 | Improved | Acc.003 | 600–800 | 105–120 | 600–1100 | 47–48 | Humera ARC |
| Setit 2 | 2016 | Improved | J-03 | 913 | 80–87 | 600–1028 | 53.77 | Humera ARC |
| Gondar 1 | 2016 | Improved | Acc.ba002 | 500–900 | 101 | 760–1022 | 50 | Gondar ARC |
| ADI | 1993 | Improved | Accadd | - | 85–91 | - | 40–58 | Werer ARC |
| Setit 3 | 2017 | Improved | J-04 | 870 | 77 | 600–1020 | 52 | Humera ARC |
| Zeri Tesfay | - | Local | - | - | - | - | - | - |
| Bounji | - | Local | - | - | | - | - | - |
| Gumero | - | Local | - | - | - | - | - | - |
| Hirhir | - | Local | - | - | - | - | - | - |

*m.a.s.l: meters above sea level

ARC: Agricultural Research Center, HU; Haramaya University

Source: [28]

## Treatment with EMS

Different concentrations of EMS, i.e. 0.0, 0.25%, 0.5%, 0.75%, and 1% were studied to find the optimum concentration [29, 30]. Seeds of 14 sesame genotypes were pre-soaked in cold tap water at 4°C for 24 hrs and then soaked in 0.5% EMS solution (HiMedia Laboratories, Pvt. Ltd., Mumbai-400086, India) with Sörenson phosphate (Sigma-aldrich, Munich, Germany) as a buffer adjusted to pH = 3 with $H_3PO_4$ (Sigma-aldrich, Munich, Germany) for 4 hrs with constant shaking at 20 rpm at 18–24°C. EMS is a mutagenic chemical, and it is important to reduce the harmful effects of EMS on the environment and human health, provided it is properly stored, used, disposed of, and transported. The procedures developed in the work of Weldemichael et al. [25] were used in this study. After washing, both the treated and control seeds were planted in well prepared beds in Kebabo, Humera and Sheraro to get M1 lines. Untreated seeds of all genotypes have been used as control for comparison throughout the experiment. Besides, no data were taken from M1, which was rather advanced into M2 (second generation mutant) lines from which all the quantitative and qualitative data were used for analyses in this study. Finally, the M2 plants derived from the M1 seeds were being considered as mutants.

## Field evaluation and recorded data of M2 lines

The experiment was laid down in a randomized complete block design (RCBD) having three replications. Each genotype of the M2 lines was randomly assigned and sown in rows, in a plot area of 2.8m by 5m with 1m between plots and 1.5m between blocks keeping inter- and intra-row spacing of 0.4m and 0.1 m, respectively. Each experimental plot was treated equally as per

**Table 2. Descriptors recorded for qualitative shoot traits of M2 lines of sesame.**

| SN | Descriptor | Scoring |
|---|---|---|
| 1 | Main stem color | 1: Green; 2: Yellow; 3: Purplish green; 4: Purple; 5: Other |
| 2 | Stem hairiness | 1: Glabrous; 2: Weak or Sparse; 3: Medium; 4: Strong or Profuse |
| 3 | Stem branch | 1: Opposite; 2: Alternate; 3: Ternate; 4: Mixed |
| 4 | Plant growth type | 1: Determinate; 2: Indeterminate |
| 5 | Branching pattern | 0: Non branching; 1: Basal branching; 2: Top branching; 3: Other |
| 6 | Stem shape in cross section | 1: Round; 2: Square |
| 7 | Leaf color | 1: Green; 2: Green with yellowish cast; 3: Green with blue-grey cast; 4: Green with purple cast; 5: Other |
| 8 | Leaf hairiness | 1: Glabrous; 2: Weak or Sparse; 3: Medium; 4: Strong or Profuse |
| 9 | Leaf arrangement | 1: Opposite; 2: Alternate; 3: Ternate; 4: Mixed |
| 10 | Leaf shape | 1: Linear; 2: Lanceolate; 3: Elliptic; 4: Ovate; 5: Narrowly cordate; 6: Other |
| 11 | Basal leaf profile | 1: Flat; 2: Cup shaped (concave); 3: Reverse cup shaped (convex) |
| 12 | Lobe incision of basal leaf | 0: Absent (leaf entire); 3: Weak; 5: Medium; 7: Strong (three or more lobes) |
| 13 | Basal leaf margin | 1: Entire; 2: Serrate; 3: Dentate |
| 14 | Leaf angle to main stem | 1: Acute (<90˚); 2: Horizontal (= 90˚); 3: Drooping (>90˚) |

Sources: [31]

the agronomic recommendations for the crop in the growing area. During the growth period, all required production practices like weed management and fertilization were carried out as recommended. Data were collected from fifteen plants of each M2 sesame lines at 75% maturity. Both quantitative and qualitative data (all in cm) were recorded properly from all the treated and control genotypes. Recorded quantitative data include: ground distance to first branch (GDFB), internode length (IL), plant height (PH), length of basal leaf (LBL), length of top leaf (LTL), width of basal leaf (WBL), width of top leaf (WTL), length of marginal leaf (LML), width of marginal leaf (WML), petiole length of basal leaf (PLBL), petiole length at middle (mid-level/mid-height) leaf (PLML), petiole length of top leaf (PLTL). Likewise, recorded qualitative data were: main stem color (MSC), stem hairiness (SH), stem branch (SB), plant growth type (PGT), branching pattern (BP), stem shape in cross section (SSCS), leaf color (LC), leaf hairiness (LH), leaf arrangement (LA), leaf shape (LS), basal leaf profile (BLP), basal leaf margin (BLM), lobe incision of basal leaf, and leaf angle to main stem (LAMS) (Table 2).

## Data analyses

The collected data for the different quantitative and qualitative traits were measured and subjected to analysis of variance (ANOVA) using GenStat 16 Software [32]. Post-ANOVA mean comparisons were carried out using Duncan's Multiple Range Test at a significance level of $p \leq 0.01$ [33].

## Results

### Effects of EMS on quantitative agronomic traits of M2 lines

Results of ANOVA clearly indicated highly significant effects among the genotypes, concentrations of EMS and their interaction effect on the quantitative data including GDFB, IL, LTL, WBL, WTL, LML, WML, and PLTL ($p \leq 0.01$; Table 3). On the other hand, the remaining

**Table 3. Analysis of variance showing mean square values and level of significance for the studied agro-morphological characters of 14 sesame genotypes.**

| SV | df | PH | GDFB | IL | LBL | WBL | LML | WML | LTL | WTL | PLBL | PLML | PLTL |
|---|---|---|---|---|---|---|---|---|---|---|---|---|---|
| | | | | | | | Traits | | | | | | |
| Block | 2 | 594.90** | 3.562 | 13.655 | 10.723 | 5.812 | 49.771 | 6.931 | 3.2243 | 0.0808 | 2.155 | 3.619 | 1.536 |
| Treatment | 1 | 22311.4** | 1085.76** | 226.714** | 0.762$^{ns}$ | 70.583** | 517.53** | 100.324** | 1.493$^{ns}$ | 37.6005** | 271.44** | 107.440** | 829.714** |
| Variety | 13 | 305.68** | 67.48** | 10.967** | 34.52** | 9.928** | 37.93** | 15.565** | 3.699** | 3.078** | 1.858$^{ns}$ | 6.939** | 6.740** |
| Treat. Var | 13 | 165.65$^{ns}$ | 13.326* | 9.278** | 6.268** | 7.744** | 12.99** | 5.506** | 6.0574** | 3.1300** | 1.030$^{ns}$ | 0.748$^{ns}$ | 6.971** |
| Residual | 54 | 91.56 | 7.106 | 2.741 | 1.702 | 1.393 | 5.049 | 2.025 | 0.8416 | 0.5210 | 1.179 | 1.335 | 1.486 |

PH: plant height; GDFB: ground distance to first branch; IL: internode length; LBL: length of basal leaf; WBL: width of basal leaf; LML: length of marginal leaf; WML: width of marginal leaf; LTL: length of top leaf; WTL: width of top leaf; PLBL: petiole length of basal leaf; PLML: petiole length at middle (mid-level/mid-height) leaf; PLTL: petiole length of top leaf.

**: $p \leq 0.01$

*: $p \leq 0.05$; ns: non-significant.

quantitative leaf traits including LBL, PH, PLBL, and PLML showed non-significant interaction effect among the genotypes and concentrations of EMS (Table 3).

The interaction effects of EMS treatment on the different quantitative traits including PH, GDFB, IL, LBL, WBL, LML, WML, LTL, WTL, PLBL, PLML, and PLTL were analyzed in this study (Table 4). Accordingly, the highest plant height was observed in the control genotypes Humera 1 (116.7 cm) and Zeri Tesfay (106.7 cm), whereas the lowest was recorded in Setit 1 (56.0 cm) and Setit 2 (52.0 cm) genotypes treated with EMS. The best ground distance to first branch was recorded in the control genotypes including Borkena (25.67 cm), Gumero (24.00 cm), Baha Necho (23.33 cm), ACC44 (22.67 cm) and Zeri Tesfay (22.67 cm). On the other hand, the treated genotypes with the least ground distance to first branch were Setit 1 (7.67 cm) and Setit 3 (8.00 cm). The highest internode length was recorded in the control genotype Setit 3 (15.33 cm) while the lowest was observed in ADI (3.00 cm), Hirhir (3.00 cm) and ACC44 (3.67 cm) genotypes treated with EMS.

The best mean LBL values were recorded in the treated genotypes of Burkena (21 cm), Bounji (16 cm), Gumero (16 cm), Humera 1 (15 cm), and Zeri Tesfay (15 cm). On the other hand, the lowest mean LBL values were observed in both the control and treated genotypes of Setit 1, Setit 2, and Setit 3 (7–10 cm). The highest mean WBL value was observed in treated genotype of Gumero (13.83 cm), while the lowest mean WBL value was recorded in the control genotype of Setit 2 (5.00 cm). The highest mean LML values were observed in the treated genotypes of Zeri Tesfay (23.00 cm), Gumero (22.67 cm), and ACC44 (21.67 cm). On the other hand, the shortest mean LML values were recorded in the control genotypes of Setit 2 (10.00 cm), Setit 1 (11.00 cm), ACC44 (11.17 cm), Borkena (11.33 cm) and Setit 3 (11.33 cm). With regard to WML, the longest mean values were recorded in the treated genotypes of Baha Necho (12.33cm) and Baha Zeyit (11.33cm), while the shortest mean values were recorded in the treated genotype Setit-2 (4.33 cm) and the control genotype Setit-1 (4.67 cm). The highest mean LTL values were observed in the treated genotype Setit 3 (8.67 cm) and control genotype Zeri Tesfay (8.00 cm). On the other hand, the lowest mean LTL was observed in the EMS treated genotypes Setit 1 (4.00 cm) and Zeri Tesfay (4.33 cm) as well as in the control genotypes Setit 1 and Setit 2 (4.40 cm and 4.33 cm, respectively). The longest mean WTL values were observed in the control Hirhir (6.33 cm), Humera 1 (5.67 cm) and Baha Zeyit (5.67 cm) as well as the EMS treated Setit 3 (6.33 cm) genotypes. On the other hand, the lowest mean WTL values were observed in the treated genotypes of Baha Zeyit, Setit 1, ADI, and Zeri Tesfay (2.07, 2.17, 2.33, 2.50 cm, respectively). The longest mean PLBL values were observed in

**Table 4. Interaction effect of EMS treatment on different quantitative agronomic traits (all in cm) of the M2 lines of sesame.**

| Genotypes | Traits | | | | | | | |
|---|---|---|---|---|---|---|---|---|
| | PH | | GDFB | | IL | | LBL | |
| | Control | Treated | Control | Treated | Control | Treated | Control | Treated |
| ACC44 | 96.7±3.33 bcd | 60.7±5.36 i-l | 22.67±2.33 ab | 13.00±1.53 g-k | 8.67±1.33 bcd | 3.67±.33 hi | 10.33±.3 f-i | 11.33±.33 e-h |
| ADI | 83.3±3.33 d-h | 72.7±5.49 f-k | 21.33±.88 a-d | 14±0.58 f-j | 9.00±1.7 bc | 3.00±.58 i | 12.33±.33 def | 9±1 hij |
| Baha Necho | 101.3±5.8 a-d | 76.3±3.33 e-i | 23.33±1.67 ab | 14.67±.88 e-i | 7.67±.33 b-g | 5.33±.67 e-i | 13±1.15 b-e | 11.33±.33 e-h |
| Baha Zeyit | 93.7±4.17 b-e | 75.7±8.17 f-j | 17.33±1.76 c-g | 9.33±.88 jkl | 7.00±.57 b-g | 4.67±.33 ghi | 11.83±1.09 efg | 11.67±.88 efg |
| Borkena | 100.0±7.64 a-d | 57.7±5.33 jkl | 25.67±.67 a | 11.67±.88 h-l | 8.00±1 b-f | 5.67±.67 d-i | 10±.57 bf-i | 10.33±.88 f-i |
| Bounji | 88.3±4.4 b-f | 66.7±6.17 h-l | 21.67±1.67 abc | 19.00±.58 b-f | 8.00±1.7 b-f | 7.67±.67 b-g | 14.33±.33 a-d | 16.33±.88 a |
| Gondar 1 | 96.3±4.48 bcd | 58.7±4.67 i-l | 19.67±1.2 b-e | 11.67±.88 h-l | 6.33±.67 b-h | 6.00±1 c-i | 13±1.53 b-e | 12.33±.33 def |
| Gumero | 102.7±7.88 abc | 59.0±8.18 i-l | 24.00±1 ab | 17.33±2.67 c-g | 7.33±1.45 b-g | 8.00±1 b-f | 11±1 e-h | 15.67±.67 a |
| Hirhir | 103.3±6.67 abc | 59.3±13.84 i-l | 16.33±.67 d-h | 13.0±1.7 g-k | 8.33±.67 b-e | 3.00±1.53 i | 10.67±1.76 e-i | 11±1.15 e-h |
| Humera 1 | 116.7±3.33 a | 72.7±8.87 f-k | 16.67±2.03 c-h | 10.0±1.15 i-l | 8.00±1.53 b-f | 4.67±.88 ghi | 13±0 b-e | 15.33±.88 ab |
| Setit 1 | 84.3±5.36 d-h | 56.0±1 kl | 12.00±3.5 h-l | 7.67±1.76 l | 8.00±1.53 b-f | 4.67±.33 ghi | 9±0 hij | 7±.58 jkl |
| Setit 2 | 87.0±2.64 c-g | 52.0±7.57 l | 14.67±.33 e-i | 10.67±1.42 i-l | 9.33±.33 b | 5.00±.58 f-i | 8.33±.33 i-l | 6.67±.33 jkl |
| Setit 3 | 105.0±2.88 abc | 72.7±3.71 f-k | 17.33±1.76 c-g | 8.00±.58 kl | 15.33±1.45 a | 6.67±.88 b-h | 9±.57 hij | 9.5±.74 ghi |
| Zeri Tesfay | 106.7±2.4 ab | 69.0±7 g-l | 22.67±1.85 ab | 14.67±.88 e-i | 8.67±1.33 bcd | 5.67±.33 d-i | 14.33±.88 a-d | 15.33±.88 ab |
| LSD | 15.66 | | 4.364 | | 2.7 | | 2.135 | |
| CV (%) | 11.8 | | 16.6 | | 24 | | 11.3 | |
| Genotypes | WBL | | LML | | WML | | LTL | |
| | Control | Treated | Control | Treated | Control | Treated | Control | Treated |
| ACC44 | 6.33±.33 f-l | 9.33±.99 b-e | 11.17±.167 gh | 21.67±4.25 ab | 6.00±1.15 f-j | 10.67±2.33 abc | 4.67±.33 ghi | 5.33±.67 e-i |
| ADI | 7.33±.33 e-k | 6.67±.88 f-l | 13.00±0 e-h | 13.33±.88 e-h | 5.83±.44 g-j | 6.67±.88 f-j | 5.67±.33 d-i | 5.33±.33 e-i |
| Baha Necho | 7.33±.67 e-k | 7.67±1.2 d-i | 13.83±1.09 e-h | 19.00±0 a-d | 8.33±.93 c-g | 12.33±.33 a | 7.33±.33 a-d | 5.33±.67 e-i |
| Baha Zeyit | 7.33±.33 e-k | 9.67±.33 bcd | 12.83±2.2 e-h | 19.33±.88 a-d | 7.00±1.04 f-j | 11.33±.33 ab | 6.33±.33 b-g | 4.67±.33 ghi |
| Borkena | 5.33±.33 i-l | 8.67±.67 b-g | 11.33±1.85 gh | 16.67±.67 cde | 6.50±.76 f-j | 8.67±1.2 b-f | 6.00±.57 c-h | 6.00±1 c-h |
| Bounji | 7.67±.33 d-i | 10.33±.88 b | 15.33±1.45 d-g | 20.00±1.15 abc | 7.67 d±.33 -h | 10.00±.577 a-e | 7.33±.33 a-d | 5.33±.33 e-i |
| Gondar 1 | 7.83±1.09 c-h | 10.00±.577 bc | 12.50±1.32 e-h | 18.00±1.15 bcd | 7.33±1.6 d-i | 10.00±.577 a-e | 7.00±1.15 a-e | 6.67±.33 b-f |
| Gumero | 6.67±.67 f-l | 13.83±.6 a | 13.17±1.59 e-h | 22.67±1.2 a | 7.17±1.01 f-i | 11.00±.577 abc | 7.67±.88 abc | 5.67±.67 d-i |
| Hirhir | 6.33±1.2 f-l | 9.33±1.76 b-e | 12.0±1.73 fgh | 16.33±2.03 c-f | 7.00±1.15 f-j | 7.00±1 f-j | 5.33±.33 e-i | 7.67 a±.67 bc |
| Humera 1 | 8.67±33 b-g | 6.33±.33 f-l | 13.33±1.45 e-h | 19.00±.58 a-d | 7.00±.57 f-j | 10.67±.33 abc | 7.00±0 a-e | 6.67±.67 b-f |
| Setit 1 | 5.67±.33 h-l | 5.67±.33 h-l | 11.00±0 gh | 12.67±1.2 e-h | 4.67±.33 ij | 5.83±.167 g-j | 4.40±.306 hi | 4.00±.58 i |
| Setit 2 | 5.00 k±0 l | 6.00±.577 h-l | 10.00±.57 h | 13.00±.58 e-h | 5.33±.33 hij | 4.33±.67 j | 4.33±.33 hi | 6.67±.88 b-f |
| Setit 3 | 6.67±.33 f-l | 7.33±.88 e-k | 11.33±.88 gh | 12.00±1.53 fgh | 7.00±.57 f-j | 6.00±.577 f-j | 5.00±0 f-i | 8.67±.33 a |
| Zeri Tesfay | 7.00±.57 f-l | 10.00±.5 bc | 16.33±2.67 c-f | 23.00±0 a | 7.07±.97 f-j | 10.00±0 a-e | 8.00±.57 ab | 4.33±.33 hi |
| LSD | 1.932 | | 3.678 | | 2.33 | | 1.5 | |
| CV (%) | 15.3 | | 14.8 | | 18.2 | | 15.3 | |
| Genotypes | WTL | | PLBL | | PLML | | PLTL | |
| | Control | Treated | Control | Treated | Control | Treated | Control | Treated |
| ACC44 | 3.833±.167 c-h | 3.00±0 e-i | 13.00±.57 ab | 8.33±.33 g | 15.33±.33 abc | 13.33±.33 c-f | 10.33±.33 def | 6.67±.33 hi |
| ADI | 4.333±.67 cde | 2.33±.33 i | 12.67±.33 abc | 8.00±.57 g | 16.0±1.150 ab | 14.00±1.15 b-f | 8.00±.57 gh | 5.33±.33 ij |
| Baha Necho | 4.00±.67 c-g | 2.83±.44 f-i | 12.00±.57 a-d | 8.00±0 g | 15.00±1 a-d | 13.00±1 d-g | 11.33±.67 de | 6.00±.577 hij |
| Baha Zeyit | 5.667±.67 ab | 2.067±.067 i | 10.67±1.2 c-f | 8.67±.33 fg | 15.00±.57 a-d | 13.00±.57 d-g | 12.00±1 cd | 4.67±.88 ij |
| Borkena | 4.00±.57 c-g | 2.667±.33 ghi | 13.00±.57 ab | 8.67±.33 fg | 16.00±.57 ab | 14.00±.57 b-f | 12.33±.88 bcd | 6.00±.577 hij |
| Bounji | 4.167±.44 c-f | 3.333±.33 c-i | 13.00±.57 ab | 9.00±0 fg | 14.67±.67 a-e | 12.67±.67 efg | 9.33±.88 efg | 4.67±.33 ij |
| Gondar 1 | 4.167±.44 c-f | 3.00±0 e-i | 12.33±.33 abc | 8.67±.33 fg | 14.33±1.2 a-f | 13.00±.577 d-g | 9.00±.57 fg | 4.33±.33 ij |
| Gumero | 4.50±.76 bcd | 3.00±0 e-i | 11.67±.88 a-d | 8.67±.33 fg | 15.00±.57 a-d | ±.57 | 11.33±.67 de | 5.67±.33 ij |
| Hirhir | 6.33±.33 a | 2.667±.67 ghi | 13.00±.57 ab | 9.33±.33 efg | 16.33±.33 a | 14.33±.33 a-f | 14.33±.88 ab | 5.00±0 ij |

*(Continued)*

**Table 4.** (Continued)

| Genotypes | Traits | | | | | | | |
|---|---|---|---|---|---|---|---|---|
| | PH | | GDFB | | IL | | LBL | |
| | Control | Treated | Control | Treated | Control | Treated | Control | Treated |
| Humera 1 | 5.667±.33[ab] | 2.667±.67[ghi] | 12.00±1.15[a-d] | 8.33±.88[g] | 16.00±.57[ab] | 14.00±.57[b-f] | 14.00±.57[abc] | 3.67±.33[j] |
| Setit 1 | 3.267±.067[d-i] | 2.167±.167[i] | 11.67±.88[a-d] | 8.67±.33[fg] | 14.00±1[b-f] | 11.00±.577[gh] | 14.67±1.45[a] | 6.00±.577[hij] |
| Setit 2 | 2.83±.167[f-i] | 3.30.351[c-i] | 13.00±1.53[ab] | 10.00±.57[d-g] | 15.00±.57[a-d] | 12.33±.33[fg] | 12.33±1.2[bcd] | 5.00±.577[ij] |
| Setit 3 | 4.667±.33[bc] | 6.33±.33[a] | 11.00±.57[b-e] | 8.67±.33[fg] | 13.67±.67[c-f] | 9.33±.33[h] | 11.00±.57[de] | 4.67±.88[ij] |
| Zeri Tesfay | 3.167±.167[d-i] | 2.50±.289[hi] | 13.33±.33[a] | 9.00±0[fg] | 16.33±.33[a] | 14.00±.577[b-f] | 11.00±.57[de] | 5.33±.88[ij] |
| LSD | 1.18 | | 1.778 | | 1.891 | | 1.996 | |
| CV (%) | 19.7 | | 10.3 | | 8.2 | | 14.6 | |

PH: plant height; GDFB: ground distance to first branch; IL: internodes length; LBL: length of basal leaf; WBL: width of basal leaf; LML: length of marginal leaf; WML: width of marginal leaf; LTL: length of top leaf; WTL: width of top leaf; PLBL: petiole length of basal leaf; PLML: petiole length at middle (mid-level/mid-height) leaf; PLTL: petiole length of top leaf.

Means followed by different letters indicate significant differences at P≤0.01, i.e., means with different letters in the column are significant, while means with the same letter(s) in the column are non-significant.

control of Zeri Tesfay (13.33 cm), Seti 2, Hirhir, Bounji, Borkena, and ACC44 (13.00 cm). On the other hand, the lowest mean PLBL values were recorded in all of the treated genotypes except Hirhir and Setit 2. The longest mean PLML values were recorded in the control genotypes of Hirhir and Zeri Tesfay (16.33 cm) followed by ADI, Borkena, and Humara 1 (16.00 cm), while the lowest mean PLML value was recorded in the treated genotype Setti 3 (9.33cm). Similarly, the control genotypes showed higher PLTL values as compared to the treated ones, the highest mean PLTL values being recorded from the control genotypes Setit 1, Hirhir, and Humera 1 (14.67, 14.33 and 14.00 cm, respectively) while the lowest being recorded from the treated Humera 1 (3.67 cm) genotype.

### Effects of EMS on qualitative agronomic traits of M2 lines

In this study, the treatment of seeds with EMS was found to be instrumental to bring major changes in multiple agronomic traits of the M2 lines of the tested Ethiopian sesame genotypes. The genotypes, EMS concentrations, and their interaction effects showed significant changes on the fourteen qualitative agronomic traits, namely main stem color, stem hairiness, stem branch, leaf shape, basal leaf profile, basal leaf margin, lobe incision of basal leaf, and leaf angle to main stem. However, the application of EMS did not result in any significant changes in plant growth type, branching pattern, stem shape in cross section, leaf color, leaf hairiness, or leaf arrangement in any of the genotypes tested (Table 5).

1. *Main stem color*. Qualitative analysis for main stem color indicated a purplish green for 100% of the treated genotypes ACC44, ADI, Gondar 1, Setit 1, Setit 2, and Setit 3. On the other hand, genotypes such as Baha Necho, Baha Zeyit, Borkena, Gumero, Hirhir, Humera 1, and Zeri Tesfay remained green whether treated with EMS or not.

2. *Stem hairiness*. The treatment of seeds with EMS caused changes in the stem hairiness from weak or sparse to medium in the genotypes ADI, Baha Necho, Baha Zeyit, and Borkena. In addition, significant change was observed in stem hairiness of Bounji, Hirhir, Humera 1, Gondar 1, Setit 1 and Zeri Tesfay genotypes from glabrous to weak/sparce. On the other hand, the application of EMS did not change the steam hairiness of the genotypes ACC44, Setit 2, and Setit 3.

**Table 5. Effect of seed treatment with EMS on different qualitative agronomic traits at M2 lines of *S. indicum* genotypes (%).**

| Leaf Morphology | Treatment | ACC44 | ADI | Baha Necho | Baha Zeyit | Borkena | Bounji | Gondar 1 | Gumero | Hirhir | Humera 1 | Setit 1 | Setit 2 | Setit 3 | Zeri Tesfay |
|---|---|---|---|---|---|---|---|---|---|---|---|---|---|---|---|
| **MSC** | | | | | | | | | | | | | | | |
| Green | Control | 100 | 100 | 100 | 100 | 100 | 100 | 100 | 100 | 100 | 100 | 100 | 100 | 100 | 100 |
| | Treated | | | 100 | 100 | 100 | 100 | | 100 | 100 | 100 | | | | 100 |
| Purplish green | Treated | 100 | 100 | | | | | 100 | | | | 100 | 100 | 100 | |
| **SH** | | | | | | | | | | | | | | | |
| Glabrous | Treated | | | | | | | 100 | | | | 100 | | | |
| | Control | | | | | | 100 | | 100 | 100 | 100 | | | | 100 |
| Weak or Sparse | Treated | 100 | | | | | 100 | | | 100 | 100 | | 100 | 100 | 100 |
| | Control | 100 | 100 | 100 | 100 | 100 | | 100 | | | | 100 | 100 | 100 | |
| Medium | Treated | | 100 | 100 | 100 | 100 | | | 100 | | | | | | |
| **SB** | | | | | | | | | | | | | | | |
| Opposite | Treated | 100 | | | 100 | 100 | | 100 | | | 100 | | | | |
| | Control | 100 | 100 | 100 | 100 | 100 | 100 | 100 | 100 | 100 | 100 | 100 | 100 | 100 | 100 |
| Alternate | Treated | | | | | | | | | | | 100 | | | |
| Ternate | Treated | | | | | | | | | | | | 100 | | 100 |
| Mixed | Treated | | | 100 | 100 | | 100 | | 100 | 100 | | | | 100 | |
| **PGT** | | | | | | | | | | | | | | | |
| Indeterminate | Treated | 100 | 100 | 100 | 100 | 100 | 100 | 100 | 100 | 100 | 100 | 100 | 100 | 100 | 100 |
| | Control | 100 | 100 | 100 | 100 | 100 | 100 | 100 | 100 | 100 | 100 | 100 | 100 | 100 | 100 |
| **BP** | | | | | | | | | | | | | | | |
| Basal branching | Treated | 100 | 100 | 100 | 100 | 100 | 100 | 100 | 100 | 100 | 100 | 100 | 100 | 100 | 100 |
| | Control | 100 | 100 | 100 | 100 | 100 | 100 | 100 | 100 | 100 | 100 | 100 | 100 | 100 | 100 |
| **SSCS** | | | | | | | | | | | | | | | |
| Square | Treated | 100 | 100 | 100 | 100 | 100 | 100 | 100 | 100 | 100 | 100 | 100 | 100 | 100 | 100 |
| | Control | 100 | 100 | 100 | 100 | 100 | 100 | 100 | 100 | 100 | 100 | 100 | 100 | 100 | 100 |
| **LC** | | | | | | | | | | | | | | | |
| Green | Treated | 100 | 100 | 100 | 100 | 100 | 100 | 100 | 100 | 100 | 100 | 100 | 100 | 100 | 100 |
| | Control | 100 | 100 | 100 | 100 | 100 | 100 | 100 | 100 | 100 | 100 | 100 | 100 | 100 | 100 |
| **LH** | | | | | | | | | | | | | | | |
| Glabrous | Treated | 100 | 100 | 100 | 100 | 100 | 100 | 100 | 100 | 100 | 100 | 100 | 100 | 100 | 100 |
| | Control | 100 | 100 | 100 | 100 | 100 | 100 | 100 | 100 | 100 | 100 | 100 | 100 | 100 | 100 |
| **LA** | | | | | | | | | | | | | | | |
| Opposite | Treated | 100 | 100 | 100 | 100 | 100 | 100 | 100 | 100 | 100 | 100 | 100 | 100 | 100 | 100 |
| | Control | 100 | 100 | 100 | 100 | 100 | 100 | 100 | 100 | 100 | 100 | 100 | 100 | 100 | 100 |
| **LS** | | | | | | | | | | | | | | | |
| Lanceolate | Treated | | 100 | | | 100 | 100 | | | 100 | | 100 | 100 | 100 | 100 |
| | Control | | 100 | | | 100 | 100 | | | | | | 100 | 100 | 100 |
| Ovate | Treated | 100 | | 100 | 100 | | | 100 | 100 | | 100 | | | | |
| | Control | 100 | | 100 | 100 | | | 100 | 100 | 100 | 100 | 100 | | | |
| **LIBL** | | | | | | | | | | | | | | | |
| Absent (leaf entire) | Control | 100 | 100 | 100 | 100 | 100 | 100 | 100 | 100 | 100 | 100 | 100 | 100 | 100 | 100 |
| Weak | Treated | 100 | 100 | | | | | 100 | | | | 100 | | 100 | |
| Medium | Treated | | | 100 | 100 | 100 | 100 | | 100 | 100 | 100 | | 100 | | 100 |
| **BLP** | | | | | | | | | | | | | | | |

(*Continued*)

**Table 5.** (Continued)

| Leaf Morphology | Treatment | Name of Genotypes | | | | | | | | | | | | | |
|---|---|---|---|---|---|---|---|---|---|---|---|---|---|---|---|
| | | ACC44 | ADI | Baha Necho | Baha Zeyit | Borkena | Bounji | Gondar 1 | Gumero | Hirhir | Humera 1 | Setit 1 | Setit 2 | Setit 3 | Zeri Tesfay |
| Flat | Treated | 100 | | 100 | 100 | 100 | 100 | 100 | 100 | | 100 | | | | 100 |
| | Control | | 100 | 100 | 100 | 100 | | 100 | 100 | 100 | 100 | 100 | 100 | 100 | 100 |
| Reverse cup shaped (convex) | Treated | | 100 | | | | | | | 100 | | 100 | 100 | 100 | |
| | Control | 100 | | | | | 100 | | | | | | | | |
| BLM | | | | | | | | | | | | | | | |
| Entire | Treated | | 100 | 100 | 100 | 100 | 100 | | | 100 | | 100 | 100 | 100 | |
| | Control | 100 | 100 | | | | 100 | | | 100 | | 100 | 100 | 100 | |
| Serrate | Treated | 100 | | | | | | 100 | 100 | | 100 | | | | 100 |
| | Control | | | 100 | 100 | 100 | | 100 | 100 | | 100 | | | | 100 |
| LAMS | | | | | | | | | | | | | | | |
| Acute (<90) | Treated | 100 | | | | | 100 | | | 100 | 100 | | | | |
| | Control | | | | | | 100 | 100 | | 100 | | | 100 | 100 | |
| Horizontal (= 90) | Treated | | | 100 | 100 | 100 | | 100 | 100 | | | | | | 100 |
| | Control | | | 100 | 100 | | | | 100 | | 100 | | | | 100 |
| Drooping (>90) | Treated | | 100 | | | | | | | | | 100 | 100 | 100 | |
| | Control | 100 | 100 | | | 100 | | | | | 100 | | | | |

MSC: main stem color; SH: stem hairiness; SB: stem branch; PGT: plant growth type; LC: leaf color; LH: leaf hairiness; LA: leaf arrangement; LS: leaf shape; LIBL: lobe incision of basal leaf; BLP: basal leaf profile; BLM: basal leaf margin; LAMS: leaf angle to main stem

3. *Stem branch*. The treatment of seeds with EMS significantly changed the stem branch in ADI, Baha Necho, Bounji, Gumero, Hirhir and Setit genotypes from opposite to mixed. On the other hand, no change was observed in the genotypes ACC44, Baha Zeyit, Borkena, Gondar 1 and Humera 1 genotypes when treated with EMS.

4. *Leaf shape*. The treatment of seeds with EMS brought changes in the shapes of the leaves in two genotypes, namely Hirhir and Setit 1, from lanceolate to ovate. On the other hand, all the other genotypes showed no change when treated with EMS.

5. *Lobe incision of basal leaf*. EMS treatment of the seeds caused changes in basal leaf profile in ADI, ACC44, Setit 1, Setit 3, and Gondar 1 genotypes from absent to weak. In addition, genotypes like Baha Necho, Baha Zeyit, Borkena, Bounji, Gumero, Hirhir, Humera 1, Setit 2 and Zeri Tesfay showed a significant change in basal leaf profile from absent to medium.

6. *Basal leaf profile*. EMS treatment brought about significant changes in basal leaf profile in ACC44, ADI, Bounji, Hirhir, Setit 1, Setit 2, and Setit 3 genotypes from flat to reverse cup shaped. The remaining genotypes showed no change when treated with the chemical mutagen.

7. *Basal leaf margin*. The treatment of EMS caused changes in basal leaf margin in ACC44, Baha Necho, Baha Zeyit and Bounji genotypes from entire to serate. The remaining genotypes, however, did not show any change when treated with the chemical.

8. *Leaf angle to main stem*. Treatment of EMS brought significant changes in the leaf angle to main stem in five genotypes. Changes from dropping to acute were observed in ACC44, Setit 2 and Setit 3; from dropping to horizontal in Borkena; and from horizontal to acute in Humera 1 genotypes.

## Discussion

Sesame (*Sesamum indicum* L.) is an important oilseed crop used for food, feed, medicinal and industrial applications. Inherently, low genetic yield potential and susceptibility to biotic and abiotic stresses contribute to low productivity in sesame. Mutation breeding has been one of the breeding strategies used to minimize yield reducing factors in sesame [34]. As a result, more than 147 sesame mutants associated with various agronomic traits, such as leaf, capsule, male sterility, flower, disease resistance, and maturity, have been reported globally [35–37]. For example, in Egypt, Cairo white 8 and Senai white 48 mutants were developed using induced mutations and released for their nonbranching habits and white seed coat color [35]. In addition, various mutant sesame varieties with increased yield, high oil content and resistance to diseases such as phytophthora blight were developed and released in India, South Korea, and Sri Lanka [34, 38, 39]. In Ethiopia, however, yield remained very low due to a diverse set of factors, including shattering capsules, determinate growth habit, branching, reduced plant height, diseases, insect pests, drought, waterlogging, salinity, and lodging. So far, no study has been conducted to improve the aforementioned desirable agronomic traits of Ethiopian sesame genotypes through mutation breeding. Therefore, improvement of sesame using mutation breeding would play a vital role to identify superior genotypes with desirable oil and yield, shattering resistance capsules, improved branching habit, determinate growth, and better number of capsules per plant. On the other hand, limitations and drawbacks of mutation breeding as a means of genetic improvement in sesame plants including low frequency, pleiotropic effects, undesirable side effects, lethal mutations, and difficult selection process would be taken in to considerations.

In this study, desirable qualitative and quantitative plant height related traits were investigated in 14 sesame genotypes using induced mutation with EMS. As a result, the lowest plant height was recorded from Setit 1 (56.0 cm) and Setit 2 (52.0) treated with EMS. These results were in line to those of Zhang et al. [11] where plant height of mutant genotypes was reduced from 170 to 110 cm. Moreover, the results of this study were consistent with previous studies that have shown a significant reduction in plant height of mutated *dw607* (*dwf1*) using EMS mutagenesis [16, 38]. According to their findings, the plant height was reduced by more than 40% (from 176.00 to 118.25 cm) as compared to the wild type, Yuzhi 11. This finding demonstrated that yield of dwarf varieties derived from *dw607* significantly increased under suitable management conditions. These findings suggest that mutation breeding is crucial for producing sesame genotypes with desirable traits, such as determinate growth habit, shattering resistant capsules, synchronous flowering, and homogeneous maturity in a short time. To the best of our knowledge, all the sesame genotypes used in this study had indeterminate growth habits, thus causing non-uniform ripening of capsules that make mechanical harvesting difficult and result in seed loss at harvest. These results agree with the findings of other studies, in which induced mutation improved desirable agronomic traits such as determinate growth in sesame [40]. According to these data, the maximum plant height acceptable for all harvest was 150 cm and lower plants were more preferable [20]. This finding has important implications for developing sesame genotypes having lower plant height. Besides, these findings suggested that higher plant height usually shows less fruiting density, is associated with shy branching, sensitive to lodging, undesirable for high yield and unsuitable for mechanical harvesting, urging more investigation for the development of dwarf sesame genotypes. As a result, reduction of plant height after seed treatment with sodium azide at low pH value has been reported [41–43]. The combination of these findings hence supports the conceptual premise that plant height decides the plant architecture and contributes a significant role in the yield of sesame [44]. In addition, dropping plant height as a means of enlightening lodging resistance is very

significant for sesame breeders. Thus, early flowering, increased yield, and reduced plant height are imperative targets for the genetic improvement of sesame.

With regard to internode length, several gammas irradiated mutant lines have been described in St. Augustine grass and in bermudagrass [45–47]. The dwarfing trait and short internode length have also been reported in the mutant *dw607* [16]. In the present study, short internode length was observed on ACC44, ADI and Hirhir genotypes treated with EMS. The findings of this study validate the findings of the previous works in identifying the role of chemical mutagens to improve genetic diversity in higher plants [48, 49]. Besides, the results of this study agree with the previous studies, which have been developed to facilitate the identification, isolation and cloning of genes used in designing crops with improved quality and yield traits in many crops such as barley, cotton, rice, peanuts, wheat, and beans [50]. These findings suggest that the observed variations in sesame genotypes are due to the induced mutations.

## Conclusion

The study analyzed the effect of EMS on improving desirable agronomic traits in M2 lines of fourteen Ethiopian sesame (*Sesamum indicum* L.) genotypes. This study showed significant changes in the majority of the qualitative and quantitative agronomic traits for the first time in Ethiopian sesame genotypes using EMS. EMS is a powerful chemical mutagen that can potentially produce desirable agronomic traits. Improving the determinate growth habit of sesame, reducing the plant height and inter node length would lead to synchronous maturity and facilitate mechanical harvesting, reducing yield losses at harvest. Further investigations to develop mutant sesame genotypes with other desirable traits such as larger seed size, improved pod shattering resistance, determinate growth habit, more uniform and shorter maturation period, modified plant architecture and size, earliness, resistance to diseases as well as higher oil content and modified fatty acid composition are strongly recommended. Moreover, the genetic variability induced in sesame plants after mutagenesis should be determined by marker assisted selection.

## Acknowledgments

The authors are highly grateful to the Tigrai Biotechnology Center Pvt. Ltd. Co. for providing laboratory facilities, Humera Agricultural Research Center for provision of seeds and field evaluation sites, and the Shire-Maytsebri Agricultural Research Center for providing filed evaluation sites. We also extend our sincere appreciation to all colleagues who contributed during the research period.

## Author Contributions

**Conceptualization:** Mohammed Mebrahtu Mossa.

**Software:** Mohammed Mebrahtu Mossa.

**Supervision:** Abraha Birhan Kassa, Yirga Belay Kindeya.

**Writing – review & editing:** Tesfaye Dissasa Bitima, Getachew Tafere Abrha, Kalkidan Tesfu, Hailay Mehari Gebremedhn.

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
