## [Decision Letter · Decision Letter 0]

4 Apr 2023

PONE-D-22-29725Improving desirable agronomic traits at M2 lines of fourteen Ethiopian Sesame (Sesamum indicum L.) genotypes using Ethyl Methanesulphonate (EMS)PLOS ONE

Dear Dr. Aabrha,

Thank you for submitting your manuscript to PLOS ONE. After careful consideration, we feel that it has merit but does not fully meet PLOS ONE’s publication criteria as it currently stands. Therefore, we invite you to submit a revised version of the manuscript that addresses the points raised during the review process.

We look forward to receiving your revised manuscript.

Kind regards,

Alia Ahmed

Academic Editor

PLOS ONE

“NO - Include this sentence at the end of your statement: The funders had no role in study design, data collection and analysis, decision to publish, or preparation of the manuscript”

4. Thank you for stating the following in the Acknowledgments/ Funding Section of your manuscript:

“The authors are highly grateful to Ethiopian Institute of Agricultural Research for funding the study, the Tigrai Biotechnology Center Pvt. Ltd. Co. for providing us facilities for laboratory, the Humera Agricultural Research Center for providing us seeds and field evaluation sites, and the Shire-Maytsebri Agricultural Research Center for giving us the filed evaluation site. We also extend our sincere appreciation to all colleagues who helped during the research period.”

“NO - Include this sentence at the end of your statement: The funders had no role in study design, data collection and analysis, decision to publish, or preparation of the manuscript”

Reviewers' comments:

Reviewer's Responses to Questions

**Comments to the Author**

1. Is the manuscript technically sound, and do the data support the conclusions?

Reviewer #1: Partly

Reviewer #2: Yes

2. Has the statistical analysis been performed appropriately and rigorously? 

Reviewer #1: Yes

Reviewer #2: No

3. Have the authors made all data underlying the findings in their manuscript fully available?

Reviewer #1: No

Reviewer #2: No

4. Is the manuscript presented in an intelligible fashion and written in standard English?

Reviewer #1: No

Reviewer #2: Yes

5. Review Comments to the Author

Reviewer #1: I revised the manuscript entitled Improving desirable agronomic traits at M2 lines of fourteen Ethiopian Sesame (Sesamum indicum L.) genotypes using Ethyl Methanesulphonate (EMS)".

The Abstract

Overall, the abstract provides a clear and concise overview of the study's purpose, methods, and results. The authors effectively highlight the importance of improving sesame yield in Ethiopia and describe how they used EMS to induce mutations in 14 sesame genotypes. They also provide a brief summary of the phenotypic changes observed in the M2 lines, and identify potential mutant genotypes that could have better desirable agronomic traits.

One strength of the abstract is that it clearly states the significance of the study, highlighting the urgent need to increase sesame yield in Ethiopia. Additionally, the authors effectively convey the novelty of their approach, noting that this is the first report on using EMS to generate desirable agronomic traits in Ethiopian sesame genotypes.

However, the abstract could benefit from additional information regarding the specific quantitative and qualitative agronomic traits that were measured and analyzed, as well as a more detailed explanation of the statistical analyses used to assess the significance of the observed phenotypic changes. Additionally, the authors could include more details about the potential implications of their findings and how they could inform future research in sesame improvement.

Finally, the abstract could be strengthened by including more specific keywords that accurately reflect the content of the study, such as "mutagenesis" and "plant breeding." Overall, the abstract provides a solid summary of the study's main findings and contributions to the field, but could benefit from more detailed information and additional keywords.

Introduction:

Overall, the introduction provides a good overview of the significance of sesame as an oilseed crop and its various uses in nutrition, industry, and medicine. However, there are some issues with the presentation of information and the structure of the text.

Firstly, the introduction could benefit from a clearer and more concise thesis statement. While the general aim of the study is mentioned towards the end of the text, a more focused research question or objective would help to guide the reader and provide context for the subsequent discussion.

Secondly, the introduction could be improved by providing more specific information about the challenges facing sesame production, such as the specific diseases and pests affecting the crop, and the current state of breeding efforts to address these challenges.

Additionally, the use of excessive citation of sources in the introduction can make the text difficult to follow and distract from the main points being made. The author should consider condensing some of the information into a more coherent narrative and relying on a few key references to support their argument.

Finally, the language and grammar in some parts of the text could be improved for clarity and precision. For example, the sentence "Therefore, it is vital to reduce plant height significantly and enhance lodging resistance so as to increase sesame production using chemical mutagenesis" could be rewritten for better clarity and readability.

Methodology section:

The material and methods section in this study appears to have some errors and critique issues. Firstly, the description of the seeds used in the study lacks information about the storage conditions, which could affect the seed quality and subsequently the results of the study. Secondly, the method of sterilization is not adequately described, specifically the duration and temperature of the sulfuric acid and glacial acetic acid treatment. This can lead to inconsistent results due to differences in sterilization efficacy.

Moreover, the section on treatment with EMS lacks information on the rationale behind the choice of EMS concentrations used, and the reason for using Sörenson phosphate buffer and H3PO4. Additionally, the method of planting the treated and untreated seeds is not clearly described, and it is unclear whether the seeds were randomly assigned to the experimental plots.

The data collection and analysis method in this study appears to be appropriate; however, there is a lack of information on the method used for assessing the level of maturity of the plants before data collection. Furthermore, the study did not include a description of the environmental conditions, which could impact the results of the experiment.

In conclusion, the material and methods section in this study has some errors and critique issues that need to be addressed for the results to be more reliable and accurate.

Results:

The study conducted ANOVA to investigate the effects of EMS treatment on quantitative and qualitative agronomic traits of M2 lines of Ethiopian sesame genotypes. The results showed significant effects of genotypes, EMS concentrations, and their interaction on the quantitative traits including ground distance to first branch, internode length, leaf length, leaf width, and petiole length. However, some quantitative traits such as plant height, petiole length of basal and middle leaves showed non-significant effects at the interaction of genotypes and EMS concentration. The study found significant variations in the mean values of different agronomic traits among the tested genotypes, with some genotypes exhibiting higher or lower values for certain traits in response to EMS treatment. However, the study did not provide a detailed discussion on the mechanisms and possible factors influencing these variations. Further studies are needed to investigate the underlying genetic and physiological mechanisms responsible for these variations and to develop effective strategies for improving the agronomic performance of sesame genotypes.

Discussion

The discussion section of this article provides an overview of the potential benefits of mutation breeding in improving the yield and desirable agronomic traits of sesame plants. However, the discussion lacks specificity and detail in certain areas. For example, while the article mentions the development and release of various mutant sesame varieties globally, it does not provide sufficient information on the specific methods and techniques used to develop these varieties. Additionally, the article does not explore the potential limitations and drawbacks of using mutation breeding as a means of genetic improvement in sesame plants.

Furthermore, the article lacks clear research objectives and hypotheses. Although the authors report on the effects of EMS treatment on plant height and internode length, they do not articulate the specific research questions that they set out to answer through their investigation. Additionally, the article does not provide a clear explanation of the significance of the results obtained from this study or how they contribute to the broader field of sesame breeding research.

Overall, the article would benefit from a more focused and structured discussion section that clearly articulates the research questions and objectives, provides a detailed analysis of the results, and explores the potential implications and limitations of the findings.

Conclusion

Overall, the conclusion of this study is well-written and provides a clear summary of the findings. However, there are a few areas where it could be improved.

Firstly, the conclusion could benefit from a more explicit statement about the significance of the findings. While it is stated that EMS is a powerful mutagen that produces desirable agronomic traits, the conclusion could be strengthened by explaining why this is important. For example, how might these findings be useful in breeding programs or in improving food security in Ethiopia?

Additionally, while the conclusion recommends further investigation and experimentation into mutant sesame genotypes, it could be more specific about what this should entail. For example, what specific traits or markers should be investigated, and what types of experiments should be conducted?

Finally, the conclusion could benefit from a brief discussion of the limitations of the study. For example, were there any factors that may have influenced the results or could have impacted the interpretation of the findings? Addressing these limitations would strengthen the conclusion by acknowledging the potential for future research to build upon these findings

Recommendation section:

There is a significant oversight in this study as it fails to include a recommendations section outlining possible interventions for future research. The absence of a recommendations section limits the usefulness and practical application of the study findings. It would be helpful to provide specific suggestions for future research on this topic, including potential areas of investigation and experimental designs that could be used to build on the current results. Additionally, without recommendations for further research, the study's conclusions may seem incomplete or insufficiently supported by evidence. It is essential to address this shortcoming by providing a detailed recommendations section that can guide future research in this area.

Reviewer #2: Improving desirable agronomic traits at M2 lines of fourteen Ethiopian Sesame (Sesamum indicum L.)

genotypes using Ethyl Methanesulphonate (EMS) seems an intersting study but it can be better if authors can add standaerror values in tables where mean values were used. Furtmore authors should write abstract in a qunatitative way by firstly writing rationale then method and results in numbers.Recommendations to all stakeholders should also be given.

6. PLOS authors have the option to publish the peer review history of their article (what does this mean?). If published, this will include your full peer review and any attached files.

Reviewer #1: **Yes: **Ahmed Abi Abdi Warsame

Reviewer #2: No

---

## [Author Response · Author response to Decision Letter 0]

2 May 2023

The authors are very hopeful that we have improved the manuscript to your satisfaction of all the points you raised.

---

## [Decision Letter · Decision Letter 1]

22 May 2023

PONE-D-22-29725R1Improving desirable agronomic traits at M2 lines of fourteen Ethiopian Sesame (Sesamum indicum L.) genotypes using Ethyl Methanesulphonate (EMS)PLOS ONE

Dear Dr. Aabrha,

Thank you for submitting your manuscript to PLOS ONE. After careful consideration, we feel that it has merit but does not fully meet PLOS ONE’s publication criteria as it currently stands. Therefore, we invite you to submit a revised version of the manuscript that addresses the points raised during the review process.

ACADEMIC EDITOR:

We look forward to receiving your revised manuscript.

Kind regards,

Alia Ahmed

Academic Editor

PLOS ONE

Journal Requirements:

Reviewers' comments:

Reviewer's Responses to Questions

**Comments to the Author**

1. If the authors have adequately addressed your comments raised in a previous round of review and you feel that this manuscript is now acceptable for publication, you may indicate that here to bypass the “Comments to the Author” section, enter your conflict of interest statement in the “Confidential to Editor” section, and submit your "Accept" recommendation.

Reviewer #1: (No Response)

Reviewer #2: All comments have been addressed

2. Is the manuscript technically sound, and do the data support the conclusions?

Reviewer #1: Partly

Reviewer #2: Yes

3. Has the statistical analysis been performed appropriately and rigorously? 

Reviewer #1: No

Reviewer #2: Yes

4. Have the authors made all data underlying the findings in their manuscript fully available?

Reviewer #1: (No Response)

Reviewer #2: Yes

5. Is the manuscript presented in an intelligible fashion and written in standard English?

Reviewer #1: No

Reviewer #2: Yes

6. Review Comments to the Author

Reviewer #1: Reviewer comment-

Based on the errors and mistakes found in the manuscript, it is clear that the paper needs significant revisions and improvements before it can be considered for publication.

Firstly, there are several spelling and grammar mistakes that need to be corrected to improve the overall readability and coherence of the manuscript. Additionally, some of the sentences are unclear or convoluted, and the authors need to revise them to improve clarity.

Moreover, the conclusion section contains some inaccuracies and unsupported claims. For instance, the authors state that "the changes observed in both qualitative and quantitative traits of this study suggested EMS to be a powerful chemical mutagen that produces desirable agronomic traits", but this conclusion is not supported by the data presented in the paper. The authors need to provide a more nuanced and evidence-based conclusion.

Finally, the authors should provide more details about their experimental design and methods, as well as their statistical analyses. They should also provide more context and background information to help readers understand the significance of their findings.

In light of these issues, the paper cannot be accepted for publication in its current form. However, if the authors are willing to make the necessary revisions and improvements, the paper could be reconsidered for publication after undergoing a thorough peer-review process.

Please look the attached file for more details

Reviewer #2: Improving desirable agronomic traits at M2 lines of fourteen Ethiopian Sesame (Sesamum indicum L.) genotypes using Ethyl Methanesulphonate (EMS) is in better shape to be further consider.

7. PLOS authors have the option to publish the peer review history of their article (what does this mean?). If published, this will include your full peer review and any attached files.

Reviewer #1: **Yes: **Ahmed Abi Abdi Warsame

Reviewer #2: No

---

## [Editor Report · Decision Letter 2]

2 Jun 2023

Improving desirable agronomic traits of M2 lines on fourteen Ethiopian Sesame (Sesamum indicum L.) genotypes using Ethyl Methane Sulphonate (EMS)

PONE-D-22-29725R2

Dear Dr. Aabrha

We’re pleased to inform you that your manuscript has been judged scientifically suitable for publication and will be formally accepted for publication once it meets all outstanding technical requirements.

Kind regards,

Alia Ahmed

Academic Editor

PLOS ONE
---

## [Editor Report · Acceptance letter]

14 Sep 2023

PONE-D-22-29725R2 

Improving desirable agronomic traits of M2 lines on fourteen Ethiopian Sesame (*Sesamum indicum* L.) genotypes using Ethyl Methane Sulphonate (EMS) 

Dear Dr. Aabrha:

I'm pleased to inform you that your manuscript has been deemed suitable for publication in PLOS ONE. Congratulations! Your manuscript is now with our production department. 

Kind regards, 

on behalf of

Dr. Alia Ahmed 

Academic Editor

PLOS ONE